

# The potential of urban rainfall monitoring with crowdsourced automatic weather stations in Amsterdam

Lotte de Vos[1,2], Hidde Leijnse[2], Aart Overeem[1,2], and Remko Uijlenhoet[1]

[1]Hydrology and Quantitative Water Management Group, Department of Environmental Sciences, Wageningen University, 6708 PB Wageningen, the Netherlands
[2]Research and Development Observations and Data Technology, Royal Netherlands Meteorological Institute, 3732 GK De Bilt, the Netherlands

*Correspondence to:* lotte.devos@wur.nl

**Abstract.** The high density of built-up areas and resulting imperviousness of the land surface makes urban areas vulnerable to extreme rainfall, which can lead to considerable damage. In order to design and manage cities to be able to deal with the growing number of extreme rainfall events, rainfall data is required at higher temporal and spatial resolutions than those needed for rural catchments. However, the density of operational rainfall monitoring networks managed by local or national authorities

is typically low in urban areas. A growing number of automatic personal weather stations (PWSs) link rainfall measurements to online platforms. Here, we examine the potential of such crowdsourced datasets for obtaining the desired resolution and quality of rainfall measurements for the capital of the Netherlands. Data from 63 stations in Amsterdam (~575 $km^2$) that measure rainfall over at least 4 months in a 17-month period are evaluated, in addition to a detailed assessment that is made of three Netatmo stations, the largest contributor of the dataset, in an experimental set-up. Although the sensor performance in

the experimental set-up and the density of the PWS-network are promising, the method of data transfer to the online platform causes considerable errors in the datasets obtained. These errors are especially large during low intensity rainfall, although they can be reduced by accumulating rainfall over longer intervals, improving the correlation with gauge-adjusted radar data from 0.48 at 5 min intervals to 0.60 at hourly intervals. Spatial rainfall correlation functions derived from PWS data show much more small-scale variability than those based on gauge-adjusted radar data and those found in similar research using dedicated

rain gauge networks. This can largely be attributed to the noise in the PWS data resulting from both the measurement setup and the data conversion by the PWS-platform. A double mass comparison with gauge-adjusted radar data shows that the median of the stations resembles the rainfall reference better than the real-time available (unadjusted) radar product. Averaging nearby raw PWS measurements already improves the match with gauge-adjusted radar data in that area. The results confirm that the growing number of internet-connected PWSs holds a promise for urban rainfall monitoring.

# 1  Introduction

Urban catchments are characterized by a high proportion of impervious surfaces, leading to a large fraction of rainfall producing direct runoff and a fast hydrological response. This makes cities especially vulnerable to flooding. The temporal and spatial resolutions of rainfall data required for urban applications exceed those needed for rural catchments (Schilling, 1991).



The rainfall information at spatial and temporal resolutions of typically 1 km by 1 km and 5 min generated by weather radar is considered valuable for urban hydrological analysis, and even forecasting (Liguori et al., 2012). However, radar has significant limitations; Rainfall is determined indirectly, over an atmospheric volume with a size depending on the distance from the radar and which may not be representative for rainfall at ground-level (Einfalt et al., 2004; Peleg et al., 2013). Errors in rainfall estimates from radar due to sampling uncertainties can be significant. In addition, there is an optimum in spatial resolution corresponding to a given temporal resolution (Fabry et al., 1994; Bell and Moore, 2000). Rain gauges, if well maintained, provide accurate ground-based measurements, although they are limited in their spatial representation. Villarini et al. (2008) showed that approximations of true rainfall with rain gauges requires a dense network and/or large temporal measurement intervals.

Hydrological models designed to deal with high resolution input have minimized rainfall accumulation errors not just when the temporal resolution or the spatial resolution is high, but particularly when the combination thereof is optimal. The required spatio-temporal resolutions for urban applications have been studied extensively. Berne et al. (2004) determined a relation between the space-time resolution required for hydrological applications as a function of the catchment size for Mediterranean conditions. It was found that for urban catchments in the order of 10 $km^2$, rainfall data is needed at a temporal resolution of 5 min and a spatial resolution of 3 km. For urban catchments of 1 $km^2$ these resolutions were 3 min and 2 km, respectively. The scales of four types of rainfall are evaluated by Emmanuel et al. (2012). With the use of variograms of 24 storm events, the required spatial resolution required to capture these types of rainfall at urban scale range from 0.8 to 3 km for instantaneous monitoring and from 2.5 to 8 km for 30 min intervals.

Gires et al. (2012) found a catchment outflow uncertainty of up to 20% due to rainfall variability at smaller scales than typical C-band radar resolution of 1 km by 1 km and 5 min. Bruni et al. (2015) address the loss in urban hydrodynamic model accuracy due to smoothing and smearing in an analysis of radar data from 4 different storms in a 3.4 $km^2$ Dutch urban catchment. Smoothing occurs when the ratio of radar resolution over catchment size becomes larger than 0.2 and storms that move near the catchment boundary are averaged partly out of the catchment. Smearing becomes significant when the ratio of the resolution of radar measurements over the rainfall correlation length exceeds 0.9, leading to averaging of rainfall over the coarse spatial grid and resulting in underestimation of rainfall rates in areas in the storm cells and overestimation in the surrounding areas. Also, a runoff peak time shift of up to 6 minutes was found due to temporal aggregation (from 1 minute to 5 and 10 minutes) of rainfall input.

Ochoa-Rodriguez et al. (2015) evaluate required spatial and temporal resolutions of rainfall in a simple spatio-temporal scaling framework. A spatial resolution of 1 km typically found in radar was found to give good hydrodynamic model results, although some extremes will be missed. Temporal resolutions should ideally be below the 5 min intervals currently available in radar-products, although 5 min radar data accuracy can be improved with the use of an accumulation procedure that assumes constant velocity of the rainfall field and rainfall intensity to vary linearly in time (Fabry et al., 1994). Coarsening temporal scales has more impact on the accuracy than coarsening spatial scales does. Initial results from an ongoing study by the authors



indicate that this impact is reduced when temporal resolutions are coarsened through aggregation (i.e. similar to rain gauges) instead of sampling. Lobligeois et al. (2014) evaluate in which circumstances hydrological model performance is enhanced by higher spatial resolution of rainfall by comparing lumped and semidistributed models with subcatchment sizes of 64, 16 and 4 km$^2$. From comparisons between the various model outputs and observations in 181 catchments in France, it was found that model accuracy improvement depends on scale, catchment and event characteristics, and that the spatial representation of rainfall can be a highly important factor in the model performance.

From these works it becomes evident that an increase of the number of measurements would yield a higher accuracy of rainfall fields and would improve hydrological applications. Adding sensors to a network is costly, although there are alternatives. For instance, rain maps can be produced from received signal strength in cellular communication networks, as the microwave signals transferred over the linkpaths are attenuated by rainfall (Overeem et al., 2016). Weather data can also be provided directly by crowdsourcing measurements from amateurs in various ways (Muller et al., 2015). A growing number of weather enthusiasts measure their local weather with automatic personal weather stations (PWSs). PWS accuracy has been evaluated for popular high-end expensive weather stations (Jenkins, 2014; Bell et al., 2015), as well as for the cheaper, user-friendly Netatmo type (Meier et al., 2015), which have grown rapidly in number over the past years. So far, weather stations have been used to obtain air temperature data to examine the Urban Heat Island effect (Steeneveld et al., 2011; Wolters and Brandsma, 2012), although other meteorological variables, such as rainfall, are measured by some of these stations as well.

A large number of PWSs share data on online platforms, both on the owner's own initiative (Gharesifard and Wehn, 2016) or automatically as an intrinsic software feature of the product (i.e. for Netatmo). Netatmo has its own online platform collecting and visualizing data from all operational Netatmo stations. The Wundermap of company Weather Underground is a similar online platform. Data from Netatmo stations are automatically linked to the Wundermap, and owners of other PWS-types can actively transmit their measurements to this platform themselves as well. A growing number of automatic weather stations are linked to these platforms; in May 2016 there were 258 personal weather stations linked to Wundermap in the Amsterdam metropolitan area (~575 km$^2$) alone (239 of type Netatmo), of which 83 stations measured rainfall (64 of type Netatmo). By contrast, the official national automatic weather station network in the Netherlands (~35,000 km$^2$) consists of 31 stations, and these are, as a rule, always located outside of urban areas. Figure 1 shows the relative resolutions in the Netherlands of networks discussed in this paper. At many locations, the density of PWS-stations collecting rainfall data far exceeds that of any realistic operational network implemented by national weather services or local authorities beyond experimental campaigns. As the online platforms collecting and sharing PWS weather data are not nation-bound, global rainfall measurements have become easily available, with especially high densities in Western-Europe, USA and Japan.

Although rainfall data availability with PWS-networks is cause for optimism for urban hydrological applications, errors are expected to be larger than those in traditional measurements. PWSs come in many types, a large fraction of which are low-cost with expected low sensor quality. In most cases there is no information available on the PWS type, the installation set-up,





maintenance of the sensor, or data post processing while transferring measurements to the online platform. Bell et al. (2013) examine the potential improvement on the UK's observational network with the real-time and local weather measurements of air temperature, relative humidity and pressure collected from Wundermap, where the most critical issue was found to be the estimation of data-quality. Validation procedures like range tests (i.e. a check whether the measurement is within predefined

extremes limits) and internal consistency tests should be applied to precipitation data from automatic weather stations (Estévez et al., 2011). Integrating crowdsourced data with variable temporal resolutions in hydrological monitoring systems by accounting for different uncertainties for data of various sources is already addressed in recent research (Mazzoleni et al., 2015).

It becomes clear that urban applications would benefit from high resolution rainfall measurements. The potential of crowd-

sourced PWS rainfall data for this purpose has not previously been explored. This study aims to determine the added value of crowdsourcing automatic weather stations for urban rainfall monitoring. For this purpose the most common PWS is tested in an experimental set-up with a high quality rain gauge reference. Additionally, a dataset of 63 crowdsourced PWS stations in Amsterdam is validated with a gridded dataset based on radar data, a manual network and a WMO certified automatic rain gauge network. These combined results provide insight in the rainfall measurement accuracy of the most commonly used PWS,

as well as any issues that occur in operational crowdsourcing of PWS rain measurements. Following this Introduction is the Methods section, where Sect. 2.1 describes the data and Sect. 2.2 gives an outline to determine the achieved measurement scales and quality of PWS, respectively. The results of an experimental PWS set-up, a comparison of a larger dataset in Amsterdam with gauge-adjusted radar data, and an analysis on inter-gauge spatial correlation of this dataset are given in Sect. 3. Finally, a Discussion on the state and future role of PWS networks in (urban) hydrological applications and Conclusions are

given in Sect. 4 and Sect. 5, respectively.

## 2   Methods

### 2.1   Data collection

#### 2.1.1   Personal weather stations

From the Wundermap website, a dataset of 63 automatic weather stations located in the Amsterdam area (~575 km$^2$) has

been retrieved. Stations were selected based on the availability of rainfall measurements, which should cover at least 4 months between December 2014 and April 2016. Of these stations, 49 are of brand Netatmo, 7 are of brand Davis, and 7 are of other unspecified brands. No details on the devices are given. According to the product specifications provided by the manufacturer, the Netatmo rain gauges have a measurement range of 0.2-150 mm h$^{-1}$ with an accuracy of 1 mm h$^{-1}$. The plastic tipping buckets have a volume of 0.1 mm and a collecting funnel with a diameter of 13 cm. The network is visualized in Fig. 2.

The Wundermap platform collects the rainfall measurements and rewrites them into rainfall over the past hour and cumulative rainfall for that day. Daily rainfall only becomes non-zero once the 0.3 mm threshold is reached and subsequent rainfall is only





reported if the rounded daily rainfall increases by at least 0.2 mm.

While Netatmo hardware can store measurements for a period of time in case of bad connectivity with the server, only real-time available data is automatically transferred to the Wundermap. This causes gaps in the Wundermap datasets where there
5 may be none in the original Netatmo data, which are only accessible to the weather station owner. Wundermap time series are therefore characterized by irregular measurement frequencies, though often 5, 10 or 15 minutes, and (large) gaps in the dataset. Also, the locations of Netatmo weather stations on the Wundermap are obtained from the settings at the Netatmo platform without notice to or confirmation from the PWS owner. Relocations of the station that are communicated to the Netatmo platform are not simultaneously adjusted on the Wundermap, leading to potentially large errors in sensor location.

We process the data obtained via Wundermap by calculating the difference in cumulative daily rainfall compared with the previous time step. Since these time steps are not fixed, this results in rainfall accumulations over time intervals of varying lengths. In order to obtain compatible time series, the rainfall is interpolated on a fixed time-line with constant steps, where constant rainfall within the original intervals is assumed. Original intervals longer than 20 min are discarded. Faulty values
in precipitation data from automatic weather stations can be identified with range tests and internal consistency tests (Estévez et al., 2011). As a first quality check, values of the interpolated time series are compared with the median rainfall of all stations for each time interval. Values exceeding this median by more than 50 mm h$^{-1}$ are excluded. Dry periods in the dataset are identified as periods of at least 24 hours where the median of all PWS measurements indicate zero-rainfall. If a PWS reports continuous zero-rainfall for at least 12 hours outside of this dry reference, the dry period is considered as faulty dry
measurements and is discarded. Finally, inter-gauge correlations are determined. If a low correlation (i.e. average and median < 0.21) is found between a station and all other stations the entire time series for that station is excluded. Visual comparison with corresponding radar rainfall time series showed that the filter that selected the data based on these criteria, was suitable in excluding obviously incorrect data from the datasets. This filter could be applied in real-time, although for operational uses outside of this dataset, adjustments are required.

**2.1.2 Radar**

As rainfall reference we use radar data from a climatological rainfall dataset by the Royal Netherlands Meteorological Institute (KNMI) (Overeem et al., 2009a, b, 2011), freely available via climate4impact.eu. This dataset is based on data from two C-band Doppler weather radars in De Bilt and Den Helder, has a temporal resolution of 5 min and a spatial resolution of 0.92 km$^2$, covering the entire land surface of the Netherlands. Radar composite images have been adjusted with rainfall measurements
from the KNMI rain gauge networks (31 automatic and 325 manual gauges). It should be noted that, due to their different representativeness, there can be significant differences between radar pixel areal rainfall and point rainfall (Schilling, 1991; Einfalt et al., 2004; Villarini et al., 2008; Peleg et al., 2013). As this radar product is adjusted with ground measurements, this difference is likely to be reduced.



### 2.1.3 Netatmo experimental set-up

As the majority of the weather stations linked to the Wundermap are of type Netatmo, we examine the quality of Netatmo rain gauges in a dedicated experimental set-up (Fig. 5, inset). As reference we use a high quality KNMI pit gauge at the Cabauw Experimental Site for Atmospheric Research (CESAR) (Leijnse et al., 2010), that measures cumulative rainfall in intervals of 12 seconds. This electronic rain gauge is placed in a so-called pit gauge configuration; a small hill of diameter 6.2 m with a circular pit with diameter 3 m and a depth of 40 cm in the middle. Precipitation is collected in the instrument (collecting funnel with a diameter of 16 cm, i.e. 200 $\mathrm{cm}^2$) and in case of solid precipitation melted by a heating element in the funnel. The amount of liquid water is measured by the position of a floating unit connected to a potentiometer. Rainfall is measured every 12 seconds within the range of 0-0.7 mm, a resolution of 0.1 mm and an accuracy of 0.2 mm. The Netatmo sensors are placed at ~40 cm around the electronic sensor in the center of the pit in such a way that the top of each sensor is level with the edges of the pit. The period considered is from February $12^{th}$ to May $25^{th}$ 2016. The datasets as collected directly from the Netatmo personal account in mm rainfall per interval of typically 5 minutes, as well as via the Wundermap platform, are compared to the pit gauge reference. One of the stations was offline between April $20^{th}$ and May $1^{st}$ and one station could not be accessed via Wunderground.

## 2.2 Analysis

### 2.2.1 Station measurement density

As mentioned previously, the original PWS data temporal resolution from Wundermap is quite irregular. Figure 3 shows time series with time steps of 5 and 10 minutes where the number of stations containing rainfall values (smoothed to daily averages) are represented. From the figure it becomes evident that the data availability is quite variable. Moreover, the fraction of the measurements over the period that is filtered out does not seem to vary significantly in time. Figure 4 shows the fraction of total pixels covering Amsterdam with a certain pixel size and time step that contain at least one measurement over the entire period. This figure shows clearly that for this dataset, increasing the frequency of measurements of the PWSs will yield a far smaller improvement in resolution than an increase in the number of measurement locations. As the current number of operational PWSs is larger than the examined dataset and growing, the data resolution from the PWS network is likely to improve significantly.

### 2.2.2 Station measurement quality

With the Netatmo experimental set-up, the performance of this type of PWS and the consequences of transferring its data to the online platform are examined. The measurements are compared to the high resolution pit gauge as well as to the radar rainfall at the corresponding pixel. These two comparisons should give an indication of differences due to sensor performance and those due to differences in representativeness of radar and rain gauges.





Rainfall measurements of the PWS dataset in Amsterdam are compared with the radar rainfall measurement at their corresponding radar pixels. Additionally, spatial correlations between stations are estimated with the use of Pearson's product-moment correlation coefficient ($r$):

$$r = \frac{E[XY] - E[X]E[Y]}{\sqrt{(E[X^2] - E[X]^2) \cdot (E[Y^2] - E[Y]^2)}} \tag{1}$$

where $E[\cdot]$ is the expectation (estimated as the arithmetic mean) and $(X, Y)$ are corresponding time series of rainfall measurements. Because of the spatial and temporal variability of rainfall, the correlation of two point-locations decreases with distance between these points. A three-parameter exponential function is suggested by Habib et al. (2001) to describe this spatial dependency relation between inter-station correlation ($r$) and distance ($d$):

$$r = r_0 \exp\left[-\left(\frac{d}{X_0}\right)^{S_0}\right] \tag{2}$$

where $r_0$ is the nugget parameter, $X_0$ is the correlation distance and $S_0$ is the shape factor. The nugget parameter $r_0$ is a measure of small scale variability and/or measurement error and is equal to 1 for perfect zero-distance correlation. Correlation distance $X_0$ indicates the distance at which the rainfall decorrelates, which should be interpreted with caution when exceeding the investigated spatial extend.

The relationship in Eq. (2) is sensitive to rainfall extremes (Habib et al., 2001), climatic regimes (Krajewski et al., 2003) and seasonality (Van de Beek et al., 2011; Tokay and Öztürk, 2012) as well as strongly dependent on time interval (Krajewski et al., 2003; Ciach and Krajewski, 2006; Van de Beek et al., 2011; Tokay and Öztürk, 2012; Van de Beek et al., 2012; Peleg et al., 2013). For the PWS-dataset in Amsterdam, correlograms are constructed and compared with spatial dependencies found in literature. Special consideration is given to the correlations between Netatmo stations as compared to the other types of rain
gauges.

## 3    Results

### 3.1    Netatmo comparison with pit gauge

The original data of three Netatmo stations (measurement frequency of ~5 min) are compared with pit gauge data (measurement frequency of 12 s) and gauge-adjusted radar data (measurement frequency of 5 min), over the period February-May 2016.
Figure 5 shows the cumulative rainfall by each, as well as scatter plots of rainfall in 10 minute intervals. The large cumulative differences in N2 and W2 as compared to the other graphs are the result of station outage, although the Netatmo stations measure less rainfall in general than the pit gauge and radar reference over this period. The scatter plots are not influenced by station outage, and show a good $r^2$ of 0.94 between Netatmo measurements and the pit gauge reference. Even though this $r^2$ suggests a small measurement error in Netatmo, the comparison with radar shows significant scatter away from the perfect fit.
This is inherent to comparisons between point locations and pixel averages, and the scatter plot resembles those reported in



Peleg et al. (2013), though the radar value used there was an average value of 12 pixels instead of 1.

The correlation between Netatmo and the electronic rain gauge is calculated for a multitude of accumulation intervals (Fig. 6). This correlation reflects small-scale rainfall variability and thus is closely related to the nugget parameter in Eq. (2). As expected, an increase of correlation is found for larger accumulation intervals. However, the correlations of data from the same devices obtained via Wundermap with the same reference shows far lower values, see Fig. 6. The original Netatmo data from the personal account has typical time steps of 5 min against 10 min for the Wundermap data. If this was the only difference between the time series, the correlation graphs should overlap for accumulation intervals above 10 min. As they only approach one another for hourly accumulations, it can be concluded that besides this effect, additional information is lost in the transfer of data between platforms.

In this study, the daily cumulative rainfall values from Wundermap are rewritten as the difference in rainfall as compared to the previous timestep. As Wundermap cumulative daily rainfall can only become non-zero when at least 0.3 mm rainfall has been collected, and later increases are only registered if they amount to at least 0.2 mm, large differences with the original time-series are caused. Especially in case of light rain, rainfall could occur for a longer period than the interval length in which the daily cumulative rainfall increases. The rainfall is then attributed to a single interval period instead of all previous intervals in which it may have been raining too. This will lead to significant differences in datasets originating from the same sensor, especially during the homogeneous light rainfall typical for Dutch winters.

Rainfall values from Netatmo stations can also be obtained with API from the Netatmo platform in daily accumulations of 10 min frequency. The same procedure as with the Wundermap platform data can be applied, which showed no such rounding occurring in this data-transfer. As cumulative rainfall can vary with steps of 0.1 mm, it is expected that more accuracy is retained. Netatmo platform datasets during April 2016 of the three experimental Netatmo stations, as well as a set of Netatmo stations in Amsterdam were examined. In these time series the time stamp of the measurements seemed to be related to the measurement collection time at the platform instead of the measurement time of the sensor, and in case of sensor outage, the last available measurement was collected repeatedly. These artifacts result in faulty interval attribution of rainfall and negatively affected the correlations with the original dataset (in case of the experimental stations) as well as with gauge-adjusted radar data.

### 3.2 Amsterdam weather station comparison with radar

Figure 7 shows the double mass plot of the filtered PWS dataset in Amsterdam, as well as the unadjusted (real-time) radar with the gauge-adjusted radar reference, where the only intervals considered are those where both time series contain measurements. Even though individual stations often do not follow the diagonal line representing a perfect match, the median of all available stations only shows a slight underestimation as compared to the gauge-adjusted radar rainfall data. This underestimation is far greater in the radar product that would be (almost) real-time available. Though large deviations occur, the median of the





stations resembles the reference quite well.

The scatter density plots in Fig. 8 show the correspondence of station rainfall against corresponding gauge-adjusted radar rainfall data over the entire period for time steps of 5, 30 and 60 minutes during periods where the radar measures non-zero

rainfall. A similar scatter as in Fig. 8 is found in Peleg et al. (2013). Not unexpectedly, the amount of scatter decreases for larger time steps. At longer accumulation intervals, the averages resemble each other more, the $CV$ decreases and the $r$ increases, indicating a better resemblance between gauge adjusted radar and station datasets.

### 3.3   Amsterdam center average comparison

In order to investigate whether the generally poor quality of individual PWS measurements can (partly) be compensated by the

generally high quantity of measurements, averages of PWS measurements are compared with radar pixel averages in a small area in Amsterdam. The selected area is the region with highest parking rates; the densely populated and touristic area of the city center and Museum square, as floods in this area will heavily impact residents, businesses and tourism alike. This region of ~20 km$^2$ is shown in Fig. 9, where the cumulative rainfall of each station relative to the mean of the 12 stations is shown. From Fig. 9 the variation between station measurements becomes evident, as well as the stations measuring highly unlikely

values considering their nearby measurements, such as station 3, 9 and 12.

The means of various subsets of the 12 PWSs are compared with the average of the 20 radar pixels over the selected Amsterdam center region. For each subset, the correlation, standard deviation and coefficient of variation of rainfall intensity is calculated over all intervals where each station contains measurements. The resulting values of each subset are represented

with boxplots in Fig. 10 per number of stations contributing to the PWS-mean. Figure 10 shows that the correlation increases and the standard deviation and CV decrease when averaging multiple stations, even when some of the station time series consist of obviously faulty measurements. By averaging the unfiltered measurements of a dozen stations, crowdsourced measurements turn out to be able to describe rainfall in the city center. As expected, the values based on 60 minute rainfall intensities show a better correspondence with gauge-adjusted radar data than 5 minute rainfall intensities.

### 3.4   Amsterdam weather station spatial correlations

Rainfall variability is often described with correlograms, see Sect. 2.2.2, describing Pearson's product-moment correlation between station pairs as a function of distance. Figure 11 shows the correlograms for the PWS-dataset, where the inner 50 percentiles of those scatter points related to inter-station correlations between Netatmo stations are indicated in red, and those between non-Netatmo stations are indicated in blue. Especially in winter (upper panels) and for short accumulation intervals,

the non-Netatmo pairs show higher correlation with one another. For longer accumulation intervals, inter-station correlations are higher and the decrease with distance is not as steep, similar to the results reported by Villarini et al. (2008), Peleg et al. (2013) and Tokay and Öztürk (2012). However, the goodness-of-fit of the correlograms differs significantly.





The correlations of all station pairs in the dataset are fitted with the relation in Eq. (2). The resulting parameters for the total dataset, as well as winter and summer individually, are given in Fig. 12. The graphs for winter show the most deviating response, suggesting irregularities in this subset in particular. The nugget parameter $r_0$ of the total dataset varies between 0.50 and 0.67 for this accumulation interval range. Villarini et al. (2008) finds a similar nugget parameter of 0.51 for 1 minute accu-

5    mulations, though far larger values at higher accumulation intervals. The nuggets found by Krajewski et al. (2003), 0.95-0.97 for 15 minutes and longer, Ciach and Krajewski (2006), 0.995 and higher for 1 minute and longer, Tokay and Öztürk (2012), 0.97 and higher for 5 minutes and longer and Peleg et al. (2013), 0.92 and higher from 1 minute and longer, are all considerably higher than the nugget parameters found here. This is unsurprising as the gauges in the networks evaluated in those papers are carefully controlled and of higher sensor quality than typical PWSs.

The correlation distance of the total PWS dataset increases with interval size in a similar manner as in previous research (Fig. 12). The erratic response of the winter graphs suggests a poor fit resulting from other factors than rainfall variability. Likely the correlation distance of stratiform winter rainfall is larger than the spatial scale examined here. The shape parameters do not seem to follow an obvious movement, similar to Peleg et al. (2013), though other research finds this parameter to increase with

interval size (Krajewski et al., 2003; Ciach and Krajewski, 2006; Villarini et al., 2008; Tokay and Öztürk, 2012).

## 4    Discussion

In the experimental set-up in Cabauw, the immediate overlying radar pixel that was first considered as reference turned out to show a significant bias as compared to gauge-adjusted radar rainfall data in all neighboring pixels. The next nearest pixel to the set-up was then used as reference instead. The distance between radar pixel center and experimental set-up thereby increased

slightly from 428.9 m to 473.5 m. Even though gauge-adjusted radar data is used as a reference, faulty measurements can occur in this dataset as well. When comparing the Amsterdam area radar pixels used in this research to their combined mean value over the 17-month period, individual time series showed up to 10% consistent higher or lower values. Biases in gauge-adjusted radar could result in a larger spread in Fig. 8, although they have a far smaller influence on the results found in Fig. 10 as the values are averaged.

Each aspect of this research, i.e. the Netatmo experimental set-up, the analysis of the station data obtained with Netatmo API and the Amsterdam PWS dataset from Wundermap, concerned time series over a different, though partly overlapping, time-period. As the shorter time series were examined with the purpose of identifying artifacts in the data, those conclusions can be carried over to the longer more robust analyses. The results on PWS data availability (Fig. 1 and Fig. 4) do not take

measurement quality into account. Because of the faulty attribution of rainfall to measurement intervals due to rounding in the data transfer, the measurements in the current form should be accumulated to larger intervals to reduce errors, although this reduces the temporal resolution appreciably. More desirable would be to address the collection method of the PWS data in the





platforms in order to maintain the quality of the original PWS rainfall measurements before data transfer.

The filter applied on the PWS dataset in this paper was based on all stations in the dataset. For operational purposes, the median value that is used as a selection criterium should be based on nearby stations only. Large rainfall values were ex-
5  cluded based on a limit on maximum rainfall of 50 mm h$^{-1}$ above the median amount of rainfall, although this potentially excludes rainfall with a return time of less than a few years (depending on the accumulation interval and the median value) (Buishand and Wijngaard, 2007). For operational purposes, faulty high values in the data should be identified without the risk of automatically excluding all extreme rainfall events. Because of the small spatial scales and the lack of extremely heavy precipitation in this dataset, the current filter was applicable, as confirmed by visual comparison with gauge-adjusted radar data.

Although a large fraction of the PWS-networks consists of Netatmo stations,this does not imply similar performance of these datasets, as factors like placement and maintenance are unknown and not necessarily equal. Even less meta-data is available on the other PWS-types in the dataset, since information on data transfer and the sensors used are not provided for those PWSs. It is expected that there is a positive correlation between the purchase costs of the PWS and the importance of maintenance and
high quality measurements to its owner, although this assumption could not be examined based on our dataset. Furthermore, the location of the station is based on the setting provided by the PWS-owner, although these may be faulty due to inaccurate localization, rounding of the longitude and latitude or relocation of the station at a later time. Even when relocations of PWSs are accurately provided to the Netatmo platform, this is not automatically communicated to Wundermap, resulting in inaccurate time series for that location. This issue is found to arise in the PWS-dataset, though the filter criterium regarding minimum
correlation with the other stations excludes time series of those stations entirely.

Different spatial correlation parameters between studies are to be expected due to different climates, rainfall types, gauge network density and -quality. However, the nugget-parameter $r_0$ found here is significantly lower than in other studies. Additionally, the nugget values of the Amsterdam dataset are significantly lower than the correlation found between the Netatmo
datasets with the electronic rain gauge reference in the experimental set-up when the data was obtained via the Wundermap platform (see also Fig. 6). This suggests the interference of additional factors besides sensor measurement errors and data transfer rounding when rainfall measurements are gathered in a less controlled manner. Such factors could be measurement errors due to station placement and poor maintenance.

It is important to note that, even though gauge-adjusted radar rainfall is used as a rainfall reference, differences with point measurements are to be expected because of representativeness errors. Ideally, a high-density gauge network could be used to improve this rainfall product in the future. A non-identical match should therefore not directly be interpreted as negative. However, as the nugget parameter from the station analyses was considerably lower than could be explained by rainfall variability alone, differences with gauge-adjusted radar data here are likely mainly caused by errors in the PWS-dataset. Besides data-
transfer errors that heavily influence the nugget parameter, the localization errors (e.g. due to shielding), that are minimized in





the experimental set-up, further decrease the nugget in the dataset analysis. When comparing nuggets from the experimental set-up and the Amsterdam dataset in the left panel in Fig. 12, the correlations found in the former do indeed reach higher values than those influenced by localization errors in the latter.

## 5    Conclusions

The resolution and quality of crowdsourced PWS rainfall measurements were analyzed to establish whether this data source allows urban hydrological applications. Although the required resolutions (as described by Schilling (1991), Berne et al. (2004), Emmanuel et al. (2012) and Ochoa-Rodriguez et al. (2015)) are not yet achieved by the current PWS-networks, the density of these networks is expected to increase. As the resolution of the current network in Amsterdam is more limited in the spatial scale than the temporal scale, the expected continued growth of PWSs that share rainfall measurements via online platforms

will yield a network approaching the desired resolutions. This offers a vast contrast compared to KNMI's automatic rain gauge network which, in the Amsterdam metropolitan area, only measures rainfall at one location outside of the city (at Schiphol airport).

From comparisons between Netatmo rainfall time series in an experimental set-up that reduces the errors due to faulty

installation to a minimum, the measurements closely resemble those from the high resolution electronic rain gauge. Larger differences are found with radar rainfall, likely due to differences in representativeness between pixels and point measurements. Although the sensor performance of this largest contributor of data in the PWS-network considered in this research looks promising, there is a significant loss in accuracy due to transfer of data to the online platform. In our analysis, this leads to attribution of rainfall at later time-intervals, which causes significant errors at small timescales. This becomes evident from

comparisons between the PWS Amsterdam dataset with gauge-adjusted radar data. The scatter density plots show large variation between datasets, especially at short time intervals. Nevertheless, the scatter density plots show more correspondence for larger accumulation intervals. Moreover, the cumulative rainfall graphs shows less systematic bias in PWS than the real-time available radar product. Averaging PWS time series further improves correlation, standard deviation and coefficient of variation with the averaged gauge-adjusted radar rainfall in a certain region (~20 km$^2$). Provided that the degree and likelihood

of overestimation of rainfall by PWSs is similar to the degree and likelihood of rainfall underestimation, as was the case in our Amsterdam city center dataset, a dense subset of PWSs can provide a good rainfall estimation over a small area, even for intervals of 5 min and without applying a quality filter.

Correlogram analyses of the PWS-dataset results in far smaller nugget-parameters than in similar research, suggesting small-

scale measurement variability not related to rainfall. Correlation fits yield especially unlikely fitted spatial correlation parameters in winter, which in the Netherlands is characterized by frequent light homogeneous rainfall events. During winter and at short intervals, the correlations between non-Netatmo type stations resembles those in literature slightly better than Netatmo-





pairs, although they are quite similar to one another for larger intervals and in summer.

The largest obstacles for the use of crowdsourced PWS datasets are the errors resulting from data-transfer, errors due to poor maintenance and faulty installations (i.e. at shielded locations). The rounding of cumulative daily rainfall measurements

occurring in the Wundermap platform and the time stamp uncertainty of measurements obtained from the Netatmo platform with API lead to considerable errors in the time series, which are only reduced at large accumulation intervals. For the purpose of a high-quality rainfall measurement network with PWS-data, these issues need to be addressed first. As the experimental set-up provided promising results regarding the sensor quality of the largest contributor to the total PWS-dataset, there is a lot to be gained from reorganizing the data transfer so that this accuracy is maintained. The resulting PWS-platform provides global

rainfall measurements that are easy to collect, located in rural areas as well as in cities, with station densities and coverage exceeding those from national weather services, and growing towards the level matching the described resolutions that are required for urban hydrological applications.

*Acknowledgements.* This research was performed as part of the RainSense project, funded by the Amsterdam Institute for Advanced Metropolitan Solutions (AMS) and the SMART city project (project number 13760) funded by Netherlands Technology Foundation (STW).

The data was made available by Weather Underground, and subsequently the weather enthusiasts sharing their weather data with the online community at the online platform Wundermap. The authors would like to thank Marcel Brinkenberg of KNMI for his assistance with the experimental set-up of the weather stations at the Cabauw Experimental Site for Atmospheric Research (CESAR). Thanks is also due to Tom de Ruijter from MeteoGroup for providing data and insight in weather measurements obtained with the Netatmo API.



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





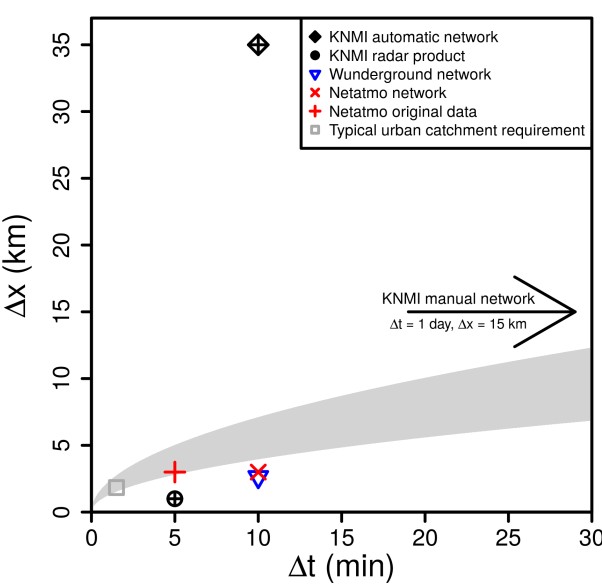

**Figure 1.** Temporal and spatial resolution of unfiltered rainfall measurements in the Netherlands with PWS network obtained via Netatmo API, Wundermap API and the potential availability of Netatmo measurements, with the resolution of KNMI's automatic and handheld rainfall measurement network. The curve represents a relation between the temporal and the spatial resolution of rainfall measurement required for urban hydrology as determined by Berne et al. (2004) for Mediterranean climate, where the square represents the value for an urban catchment with surface area of 0.1 km$^2$.





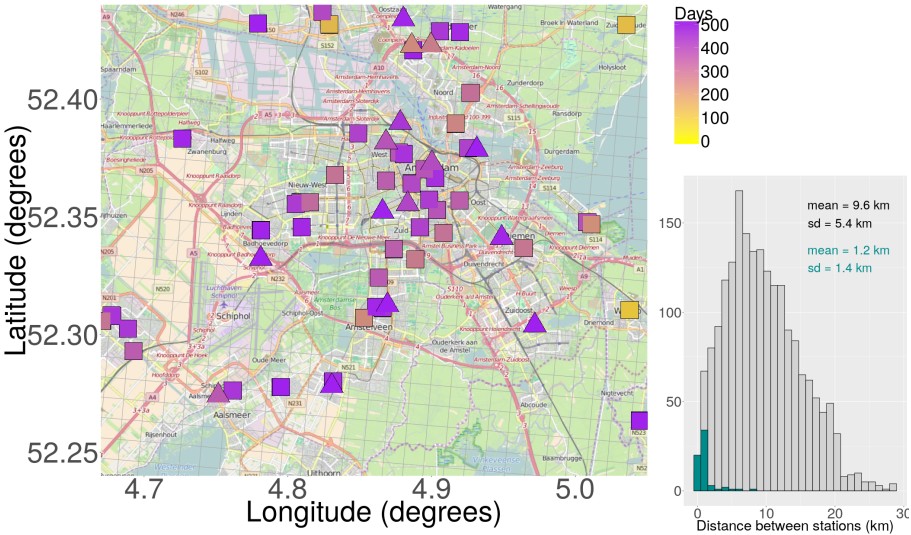

**Figure 2.** Locations and operational days (i.e. days with measurements) of Netatmo (squares) and other types (triangles) of PWSs, with the radar pixel grid in the Amsterdam metropolitan area. Inter-station distances are represented in the histogram, colored green for nearest neighbor distances.

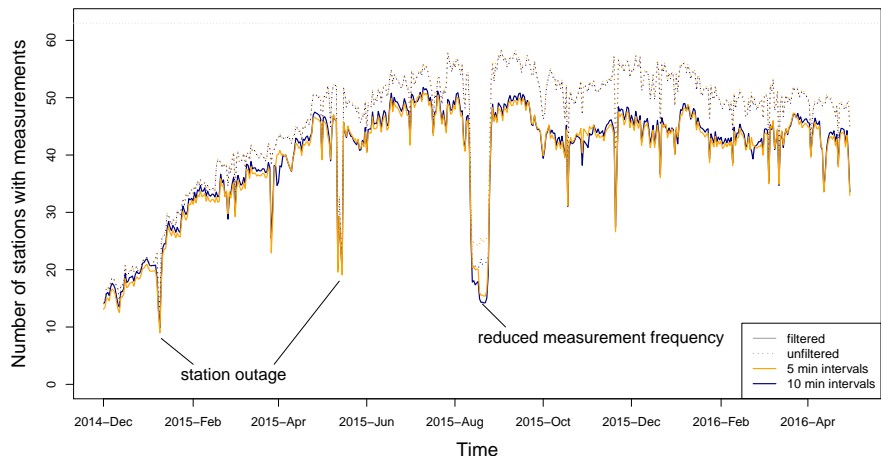

**Figure 3.** Number of stations with rainfall data from the PWS dataset, before and after applying filter, for every 5 and 10 minute interval over the entire period, smoothed per day. The two indicated dips corresponds to complete outage of stations, the third with a longer period of fewer measurements in all stations.





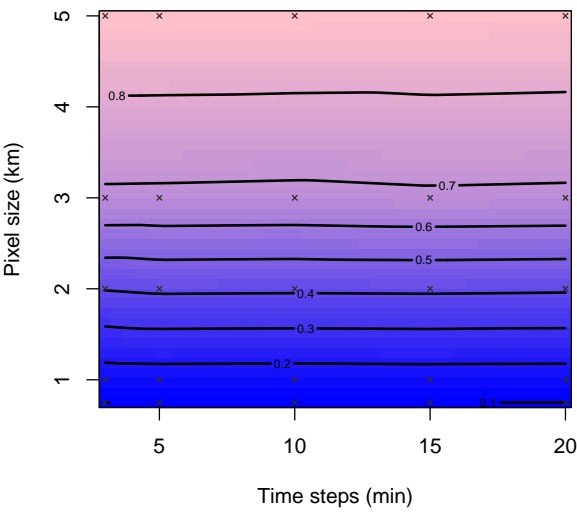

**Figure 4.** Fraction of pixels containing at least one measurement (after filter is applied) for various combinations of time steps and pixel grid size (represented with cross) over the Amsterdam area between December 2014 and April 2016.

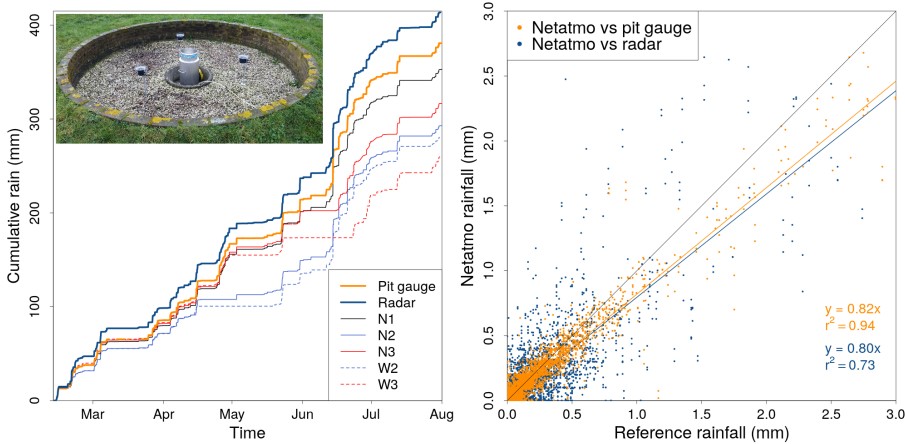

**Figure 5.** Left panel: Cumulative rainfall according to reference pit gauge, gauge-adjusted radar, Netatmo stations (N1, N2 and N3) and Netatmo stations obtained via Wundermap (W2 and W3). N2 (and consequentially W2) was offline between April $20^{th}$ and May $1^{th}$. Photo shows the experimental set-up of the rain gauges in the pit gauge configuration. Right panel: scatter plots of 10 minute rainfall and linear fits of rainfall according to N1, N2 and N3 as compared to reference pit gauge (orange) and radar (blue).





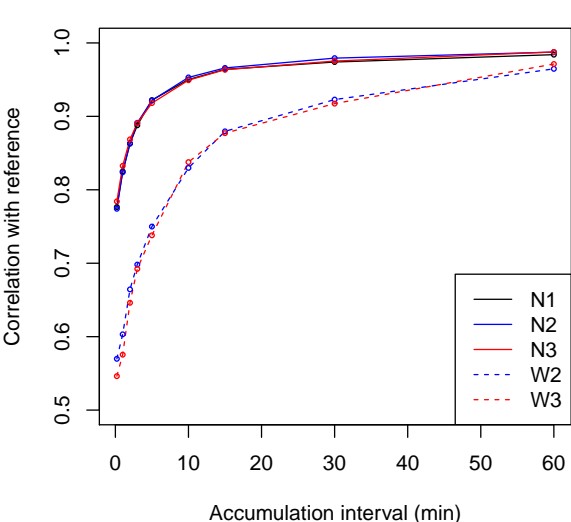

**Figure 6.** Correlation between rainfall measurements by Netatmo stations (N1, N2 and N3), also obtained via Wundermap (W2 and W3), and the pit gauge reference for various accumulation time steps.





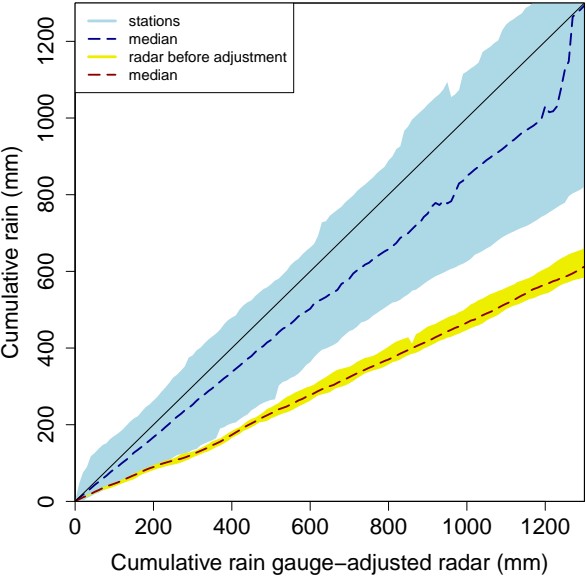

**Figure 7.** Double mass plots of station filtered rainfall measurements and real-time available radar data with gauge-adjusted radar rainfall at the corresponding location in the period between December 2014 and March 2016. Only intervals where both radar and station contain measurements are taken into account. Colored regions indicate the range between minimum and maximum of the graphs and dashed lines represent the median of the combined datasets.

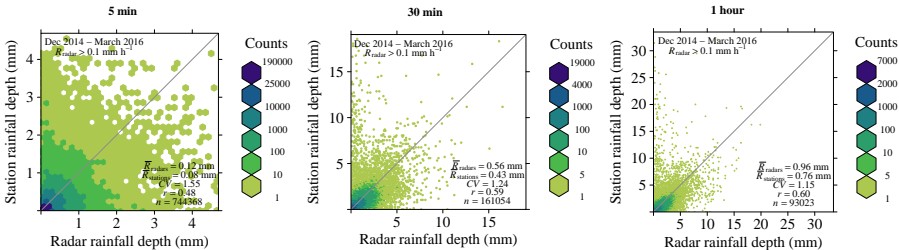

**Figure 8.** Scatter density plots of all station rainfall measurements against the gauge-adjusted radar rainfall data in the corresponding radar pixel when radar reported non-zero rainfall ($> 0.1$ mm). The $\bar{R}_{\text{radars}}$, $\bar{R}_{\text{stations}}$, $CV$, $r$ and $n$ values in the panels represent the average rainfall according to the gauge-adjusted radar data, the average rainfall according to the stations, the coefficient of variation, the correlation and the number of intervals respectively. Graphs are made for 5 min, 30 min and hourly accumulation intervals.





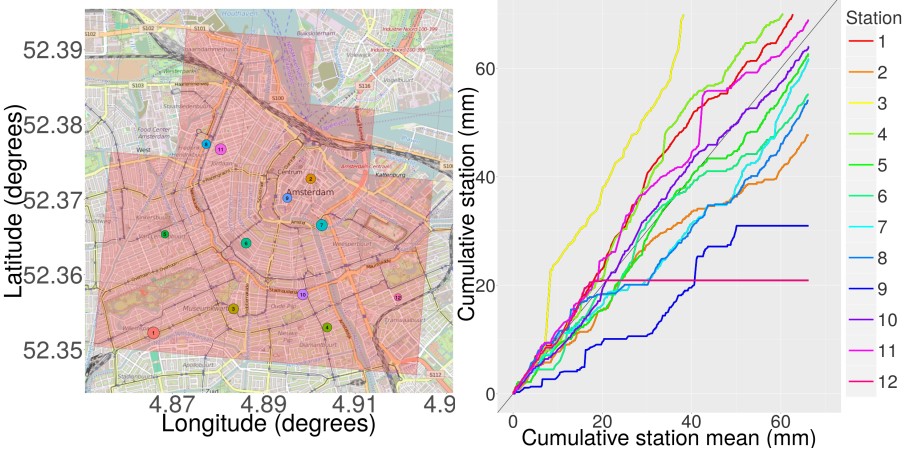

**Figure 9.** Left panel: Locations of 12 stations and 20 radar pixels in the city center of Amsterdam, where symbol size represents the number of unique days with measurements by the station (range 371-514 days). Right panel: Double mass plots of the station measurements as compared to the mean of all 12 stations over the intervals where all stations contain measurements.

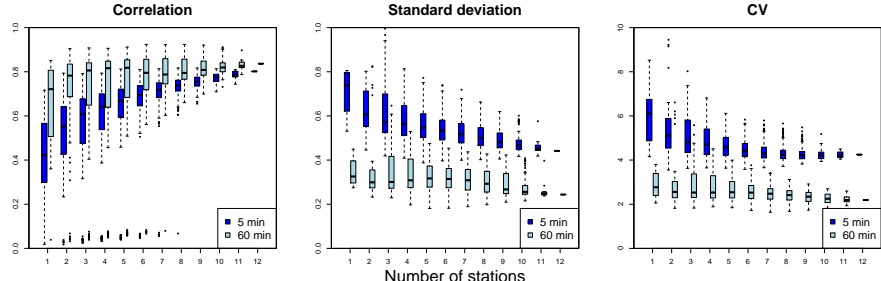

**Figure 10.** Boxplots of correlation, standard deviation and coefficient of variation (CV) of rainfall intensity for all subset means within the 12 stations in Amsterdam city center, as compared to the radar rainfall intensity 20 pixel mean for interpolated 5 minute and hourly time series.





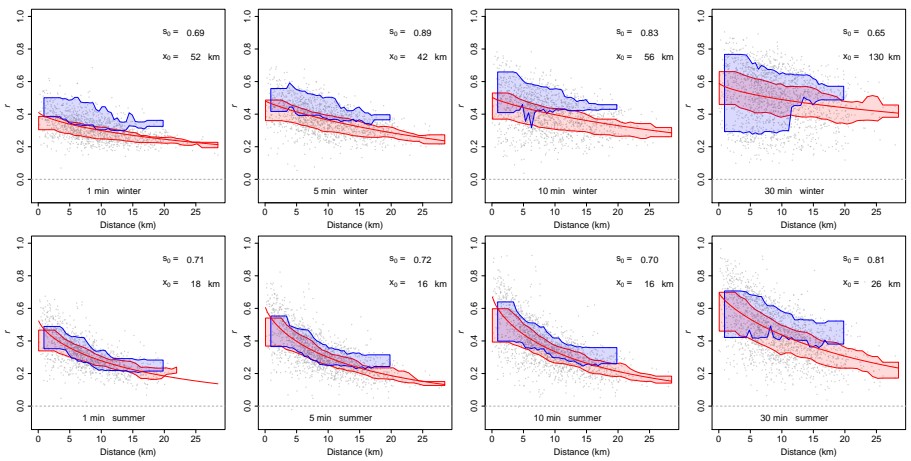

**Figure 11.** Correlograms of all stations after filtering at various accumulation values for winter (top panels) and summer (bottom pannels). The red and blue areas represent the interquartile range of the Netatmo stations and non-Netatmo stations respectively. The areas are constructed with a moving window of width 5 km. The scatter plots are fitted with the exponential relation stated in Eq. (2), which fitting parameters are given in the panels.

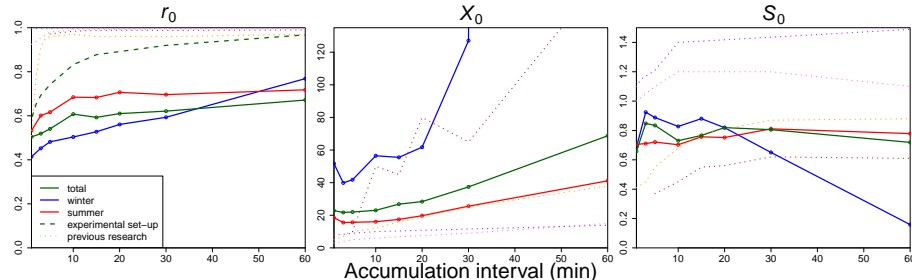

**Figure 12.** Timescale dependency of nugget (left panel), correlation (middle panel) and shape (right panel) parameters from fit described in Eq. (2), for total PWS dataset, as well as winter and summer only. Dotted lines represent values found in previous research by Peleg et al. (2013) (violet), Villarini et al. (2008) (orange), Tokay and Öztürk (2012) (brown) and Ciach and Krajewski (2006) (purple), where the dashed line in the first panel shows the timescale dependency of the Netatmo station nugget found in the experimental set-up as previously shown in Fig. 6.