# Peer review of "The potential of urban rainfall monitoring with crowdsourced automatic weather stations in Amsterdam"

_Hydrology and Earth System Sciences, 2016_

## Referee Comment (RC1) · Anonymous Referee #1 · 31 Oct 2016

The paper discusses the advantages of crowdsourced weather station data (rainfall measurement) to obtain rainfall information suitable for hydrology studies in urban areas, i.e., rainfall measurements that the need to have high temporal and spatial resolutions. The paper is, to the best of my knowledge, the first attempt to quantify the errors of rainfall data made available from local, distributed and crowdsourced weather stations, which makes it an interesting study. In the paper the crowdsourced rainfall data are compared with dedicated rain gauges and rainfall radar data as these are the common rainfall data sources used in urban hydrology.

Here some suggestions: (1) Some sentences are too vague and need to be rephrased to convey a clear message: e.g., what do authors mean by "... return time of less than a few years..."? (2) The structure of the manuscript deserves to be revised. See for example: (a) the order figure numbers appear in the manuscript is cumbersome and

makes the manuscript confusing (why Fig. 5 appears earlier than Fig. 3?) (3) Lines 18-24 in Page 12 are not conclusions. The authors may want to move these sentences to another section of the manuscript. Also, lines 29-33 page 12 are not conclusions. (4) Figure 5 does not show ". . . a dedicated experimental set-up . . ." (page 6, lines 2-3), i.e. the text does not match what is seen in the Figure. The authors may want to adjust the text of the Figure. (5) Figure 1. The "black dot" KNMI radar product is not visible in the plot (only in the legend). Authors may want to adjust the plot /or legend). (6) There is room for improving the English language; incomplete sentences (e.g., Page 5, line 6) and minor typos (e.g., "criterium" should read "criterion" in Page 11, line 19) can be found in the manuscript; "CV" is only defined in Fig. 10 legend.

---

## Author Comment (AC1) · 3 Nov 2016

*REVIEWER: The paper discusses the advantages of crowdsourced weather station data (rainfall measurement) to obtain rainfall information suitable for hydrology studies in urban areas, i.e., rainfall measurements that the need to have high temporal and spatial resolutions. The paper is, to the best of my knowledge, the first attempt to quantify the errors of rainfall data made available from local, distributed and crowdsourced weather stations, which makes it an interesting study. In the paper the crowdsourced rainfall data are compared with dedicated rain gauges and rainfall radar data as these are the common rainfall data sources used in urban hydrology.*

AUTHORS: We thank the reviewer for the valuable review of this paper. We appreciate

the constructive comments, and address each of them below.

*REVIEWER: Here some suggestions:*

*(1) Some sentences are too vague and need to be rephrased to convey a clear message: e.g., what do authors mean by "... return time of less than a few years"?*

AUTHORS: The section discussed here was intended to highlight the limitations of the filter that was used on the Amsterdam dataset. "a return time of less than a few years" refers to the return times of Dutch rainfall amounts found by Buishand and Velds (1980), Overeem et al. (2008) and Buishand and Wijngaard (2007), as documented by the latter.

The filter that is discussed in this section contains a dynamic maximum threshold, which filters out intervals with values higher than 50 mm h$^{-1}$ above the median rainfall intensity of the surrounding stations. For example, in case of 10 min time series the filter works as follows; If for a certain 10 min interval the median rain intensity of surrounding stations is 4 mm h$^{-1}$, the upper threshold for rainfall intensity becomes 54 mm h$^{-1}$ for this particular interval. According to the values found in the research, this corresponds with a return time between 1 and 2 years (Buishand Velds, 1980; Buishand Wijngaard, 2007).

As these return time values are different for other interval lengths and median values of surrounding stations, the statement was phrased in a general way.

In conclusion, realistic rainfall measurements could be excluded by the filter. We would like to stress that visual comparison with gauge adjusted radar data indicates that this was not the case for the dataset used is this paper. However, for operational application of a filter with an upper threshold, this should be taken into consideration.

Other sentences will be reevaluated for vagueness.

*(2) The structure of the manuscript deserves to be revised. See for example: (a) the order figure numbers appear in the manuscript is cumbersome and makes the manuscript confusing (why Fig. 5 appears earlier than Fig. 3?)*

AUTHORS: In Sect. 2.1.3 the experimental set-up is described, and the reader is referred to the inset in Fig. 5 where the set-up is visualized. This reference precedes references to Fig. 3 and Fig. 4, as the results that are also visualized in Fig. 5 are discussed in Sect. 3.1. The authors acknowledge this may be cumbersome to the reader, though prefer to not split up Fig. 5 in two figures.

*(3) Lines 18-24 in Page 12 are not conclusions. The authors may want to move these sentences to another section of the manuscript. Also, lines 29-33 page 12 are not conclusions.*

AUTHORS: The authors agree these sentences should be moved to the results section.

*(4) Figure 5 does not show "a dedicated experimental set-up" (page 6, lines 2-3), i.e. the text does not match what is seen in the Figure. The authors may want to adjust the text of the Figure.*

AUTHORS: With this sentence the authors refer to the photo in Fig. 5 that shows the experimental set-up of the rain gauges. This will be rephrased in the text.

*(5) Figure 1. The "black dot" KNMI radar product is not visible in the plot (only in the legend). Authors may want to adjust the plot /or legend).*

AUTHORS: The symbols in Fig. 1 will be adjusted in the revised version.

*(6) There is room for improving the English language; incomplete sentences (e.g., Page 5, line 6) and minor typos (e.g., "criterium" should read "criterion" in Page 11, line 19) can be found in the manuscript; "CV" is only defined in Fig. 10 legend.*

AUTHORS: The overall English language will be reevaluated, and your suggestions will be implemented.

References:

Buishand, T. A. and Velds, C. A.: Neerslag en verdamping [Precipitation and evaporation], Royal Netherlands Meteorologic Institute, 1980.

Buishand, T. A. and Wijngaard, J.: Statistiek van extreme neerslag voor korte neerslagduren [Statistics of extreme rainfall for short durations], Royal Netherlands Meteorologic Institute, 2007.

Overeem, A., Buishand, A. and Holleman, I.: Rainfall depth-duration-frequency curves and their uncertainties. Journal of Hydrology, 348(1), 124-134, 2008.
* * *

---

## Referee Comment (RC2) · Anonymous Referee #2 · 15 Nov 2016

Summary

This study discusses a new technique for measuring rainfall. This consequently would satisfy the need for hydrological analyses like urban hydrology, where high temporal and spatial resolution rainfall data are required. Although several studies have addressed alternative ways for measuring rainfall such as using microwave links, this study specifically investigates rainfall information from automatic personal weather stations (PWSs) for point-measurement purposes. I found the topic quite interesting as such approaches may indeed provide valuable information.

General comments:

I found the manuscript a bit hard to follow. The authors may consider the following points for improvements. The structure of the manuscript may need some modifica-

tions. For example, you may introduce the "Methodology" the "Data and study area" in separated sections, and not combined. Providing some important details would make getting the message of the manuscript easier. More information is required explaining the functionality of the devices used in this study. Although some devices do not have the sought information, providing available information, e.g. for Netatmo, would help to better follow the text. You may even add some figures for that reason. The message of some sentences is not clear. Furthermore, some sentences have minor/major language problems and/or they are too long. You may consider English proofreading. Another point is the objective of this study. The last paragraph in introduction addresses the objective of this study. You may clearly add the fact that using these measurements is economically reasonable comparing to conventional techniques. At the end I would like to add that there are some facts given without proper references.

Scientific comments:

You addressed a study where they used "average value of 12 pixels", and furthermore, you addressed the difficulties radar data have, especially when comparing to "ground-based measurements". You used in this study 1 radar pixel as the reference at the end. You may elucidate the reason for that. Also, you may explain the method (you) used for correcting (adjusting) radar data.

The same is valid for the experimental set-up. You may add how exactly the reference device "KNMI pit gauge" measures rainfall.

I could not really understand the reason why you used Pearson correlation coefficient for your analyses. You may add some other criteria like bias, root mean square error, etc. Regarding the "inter-station correlation", you may add the way you estimated the parameters.

Specific comments:

You may address the specific comments by considering the points mentioned above.

Some of the specific comments for the abstract and introduction section is provided in the following:

P.1 - L.7: are you referring by "63 stations" to the Netatmo network in Amsterdam?

P.1 – L.10: "the method of data transfer to the online platform causes considerable errors in the datasets obtained." This phrase may need some modifications.

P.1 – L.16: Does "data conversion" refer to "data transfer" you provided some lines earlier? The word "conversion" sounds a bit strange in this contest.

P.2 – L.3: Why "Rainfall" starts with capital letter? And "from the radar"; you may change to "from the radar station".

P.2 – L.4: "... and which may.." I suppose one may not put "and" and "which" together.

P.2 – L.7: Does "spatial resolution" mean "network density"?

P.2 – L.10: You may add a comma after "Hydrological models" and a comma before "have minimized".

P.2 – L.15: What is the spatial resolution of 3 km? Is it a 3 km $\times$ 3 km ?

P.2 – L.21: Bruni et al. (2015) "addressed" ….

P.2 – L.24: When explaining "Smearing"; "…the ratio of the resolution…": are you referring to temporal, spatial or tempo-spatial resolution?

P.2 – L.30: Ochoa-Rodriguez et al. (2015) "evaluated the …"

P.2 – L.31: You may add a comma after "1 km" and a comma before "was found".

P.2 – L.34: Not all radar products are in 5 min temporal resolutions. Some X-Band radars work in 30-sec temporal resolution.

P.2 – L.35: "… impact on the accuracy than coarsening spatial scales does." I would change the word scale to "resolution". Furthermore, you may omit "does" at the end of
the sentence.

P.3 – L.2: "... evaluate" to "... evaluated". You may rephrase the entire sentence.

P.3 – L.9: What type of sensors are you referring to? Rain gauges?

P.4 – L.10: But you did not investigate for any urban applications. Your study area is located in an urban area.

---

## Author Comment (AC2) · 22 Nov 2016

*REVIEWER: Summary*

*This study discusses a new technique for measuring rainfall. This consequently would satisfy the need for hydrological analyses like urban hydrology, where high temporal and spatial resolution rainfall data are required. Although several studies have addressed alternative ways for measuring rainfall such as using microwave links, this study specifically investigates rainfall information from automatic personal weather stations (PWSs) for point-measurement purposes. I found the topic quite interesting as such approaches may indeed provide valuable information.*

[Figure]

AUTHORS: We want to thank the reviewer for the review of this paper. We appreciate all the suggestions to improve this manuscript, and address each of them below.

*REVIEWER: General comments:*

*I found the manuscript a bit hard to follow. The authors may consider the following points for improvements. The structure of the manuscript may need some modifications. For example, you may introduce the "Methodology" the "Data and study area" in separated sections, and not combined. Providing some important details would make getting the message of the manuscript easier. More information is required explaining the functionality of the devices used in this study. Although some devices do not have the sought information, providing available information, e.g. for Netatmo, would help to better follow the text. You may even add some figures for that reason. The message of some sentences is not clear. Furthermore, some sentences have minor/major language problems and/or they are too long. You may consider English proofreading. Another point is the objective of this study. The last paragraph in introduction addresses the objective of this study. You may clearly add the fact that using these measurements is economically reasonable comparing to conventional techniques. At the end I would like to add that there are some facts given without proper references.*

AUTHORS: We thank the reviewer for the suggestions on how to improve the manuscript, and we will re-evaluate the structure of the sections in order to improve readability. The text will be scanned for sentences that are unclear and/or without proper reference. The argument of the reviewer regarding the objective of this study will be added to the last paragraph of the Introduction.
Section 2.1.1. contains product specifications (as provided by the manufacturer) on the Netatmo rain gauge devices. More information could be given on the data transfer between devices in a station; the rain module, i.e. the tipping gauge, communicates in a wireless manner to the indoor module over distances up to 100 m. Every 5 minutes (time step may vary) the number of tips in that interval is communicated from the indoor module to the online dashboard via a WiFi-connection. Such background information will be discussed more prominently in this section.

*REVIEWER: Scientific comments:*

*You addressed a study where they used "average value of 12 pixels", and furthermore, you addressed the difficulties radar data have, especially when comparing to "ground-based measurements". You used in this study 1 radar pixel as the reference at the end. You may elucidate the reason for that.*

AUTHORS: Though radar provides rainfall estimates with a large coverage, they are indirect measurements of rainfall averages over a spatial area (pixels 0.92 km$^2$ for our C-band Doppler radar product) representative for a significant altitude above the ground ($\approx$ 1.5 km). Because radar measures in a volume aloft, there will always be differences with point measurements at the ground. This difference will not decrease by averaging rainfall in multiple radar pixels. Although averaging over radar pixels may smooth such an error, it also implies that rainfall from a much larger area is compared to a point measurement, which will introduce an additional difference. Figure 5 shows the scatter plots of Netatmo measurements with radar data as well as with the pit

gauge reference. This figure is meant to highlight that even though the Netatmo rain gauges show good correspondence with the pit gauge reference (both are point measurements), a larger spread is found when comparing with gauge-adjusted radar rainfall. This is helpful in the interpretation of the analyses of PWS rainfall data in Amsterdam. For these time-series no ground-based point reference is available, which is why we compare with the gauge-adjusted radar data.

*REVIEWER: Also, you may explain the method (you) used for correcting (adjusting) radar data.*

AUTHORS: Section 2.1.2. describes the radar dataset used in this research. Data from two C-band Doppler weather radars in De Bilt and Den Helder have been adjusted with ground-based rainfall measurements. This adjustment was already done and has not been performed as part of this study. The dataset has been made available by KNMI. For information on the methods used we refer to previous research (Overeem et al., 2009a, b, 2011), though some more background will be provided in Sec. 2.1.2.

*REVIEWER: The same is valid for the experimental set-up. You may add how exactly the reference device "KNMI pit gauge" measures rainfall.*

AUTHORS: The KNMI pit gauge has been introduced in Sect. 2.1.3., which also includes information on the pit gauge configuration. It is an electronic rain gauge, measuring cumulative rainfall every 12 s. A potentiometer connected to a floating unit measures the amount of liquid water. A heating element makes it possible for the device to accurately measure solid precipitation.

*REVIEWER: I could not really understand the reason why you used Pearson correlation coefficient for your analyses. You may add some other criteria like bias, root mean square error, etc.*

AUTHORS: The Pearson correlation coefficient is a widely used statistic to describe correspondence between rain gauge measurements, in order to characterize the complex spatial structure of rainfall patterns. The dependence of the Pearson correlation coefficient with distance between gauges has been evaluated by Habib et al. (2001), Krajewski et al. (2003), Ciach and Krajewski (2006), Villarini et al. (2008), Tokay and Öztürk (2012) and Peleg et al. (2013). In order to compare with results in these papers, a similar method was chosen.

Figure 6 describes the accumulation interval dependence of the correlation between gauges and the pit gauge reference. The same figure has been made for the coefficient of variation (CV) of the residuals. The CV values decrease with increasing accumulation interval where they approach zero (indicating a perfect match) at 15 minute intervals. The values of the Wunderground timeseries approach zero at a slower pace than the Netatmo original data. As this CV-analysis did not provide any additional information beyond that communicated by Fig. 6, it has not been included in the manuscript.

The Pearson correlation coefficient and the CV give information on the random error. Figure 5 and Fig. 7 give information on the bias.

*REVIEWER: Regarding the "inter-station correlation", you may add the way you estimated the parameters.*

AUTHORS: The parameters in Fig. 12 have been determined by fitting Eq. (2), where the correlations between stations and the inter-gauge distances were used as input. Fitting was done by determining the nonlinear (weighted) least-squares estimates of the parameters. This will be included in Sect. 3.4.

*REVIEWER: Some of the specific comments for the abstract and introduction section is provided in the following:*

*P.1 - L.7: are you referring by "63 stations" to the Netatmo network in Amsterdam?*

AUTHORS: The 63 stations do not exclusively refer to Netatmo stations. Figure 2 shows which of these stations are of type Netatmo (square symbols) and which are of other types (triangle symbols). The time-series are obtained from the Wundermap website, as described in detail in Sect 2.1.1. We will clarify this in the revised manuscript.

*REVIEWER: P.1 – L.10: "the method of data transfer to the online platform causes considerable errors in the datasets obtained." This phrase may need some modifications.*

AUTHORS: The sentence will be rephrased in "The sensor performance in the experimental set-up and the density of the PWS-network are promising. However, features in the online platforms cause changes in the time-series, resulting in considerable errors in the datasets obtained."

*REVIEWER: P.1 – L.16: Does "data conversion" refer to "data transfer" you provided some lines earlier? The word "conversion" sounds a bit strange in this contest.*

AUTHORS: This refers to adjustments made to the measurement time-series by features of the online platforms. Data conversion occurs within the data transfer. This will be rephrased in order to avoid confusion.

*REVIEWER: P.2 – L.3: Why "Rainfall" starts with capital letter? And "from the radar"; you may change to "from the radar station".*

AUTHORS: This should indeed not be a capital. "from the radar" will be changed into "from the radar station" as suggested.

*REVIEWER: P.2 – L.4: "and which.." I suppose one may not put "and" and "which" together.*

AUTHORS: This will be corrected.

*REVIEWER: P.2 – L.7: Does "spatial resolution" mean "network density"?*

AUTHORS: Spatial resolution is directly related to network density, only if all stations in the network are measuring constantly. In most PWS-networks, spatial resolution is lower than the actual network density because of station outages and periods of fewer measurements in time.

*REVIEWER: P.2 – L.10: You may add a comma after "Hydrological models" and a comma before "have minimized".*

AUTHORS: This will be corrected.

*REVIEWER: P.2 – L.15: What is the spatial resolution of 3 km? Is it a 3 km x 3 km?*

AUTHORS: A spatial resolution of 3 km generally corresponds with a 3 km x 3 km square. Here it means that measurements are taken on average 3 km apart. For crowdsourced measurements, the inter-gauge distance is likely to vary slightly, as the placement of stations is irregular and not in a regular grid.

*REVIEWER: P.2 – L.21: Bruni et al. (2015) "addressed".*

AUTHORS: This will be corrected.

*REVIEWER: P.2 – L.24: When explaining "Smearing"; "the ratio of the resolution": are you referring to temporal, spatial or tempo-spatial resolution?*

AUTHORS: Here we refer to spatial resolution ratios. This will be clarified in the text.

*REVIEWER: P.2 – L.30: Ochoa-Rodriguez et al. (2015) "evaluated the"*

AUTHORS: This will be corrected.

*REVIEWER: P.2 – L.31: You may add a comma after "1 km" and a comma before "was found".*

AUTHORS: This will be corrected.

*REVIEWER: P.2 – L.34: Not all radar products are in 5 min temporal resolutions. Some X-Band radars work in 30-sec temporal resolution.*

AUTHORS: The phrase will be changed to "Temporal resolutions should ideally be below the 5 min intervals currently available in most operational weather radar-products,".

*REVIEWER: P.2 – L.35: "impact on the accuracy than coarsening spatial scales does." I would change the word scale to "resolution". Furthermore, you may omit "does" at the end of the sentence.*

AUTHORS: This will be corrected.

*REVIEWER: P.3 – L.2: "evaluate" to "evaluated". You may rephrase the entire sentence.*

AUTHORS: This will be corrected.

*REVIEWER: P.3 – L.9: What type of sensors are you referring to? Rain gauges?*

AUTHORS: The sensors we refer to are any type of sensors that measure rainfall. We

will clarify this in our revised manuscript. This could be rain gauges, though it could also refer to disdrometers.

*REVIEWER: P.4 – L.10: But you did not investigate for any urban applications. Your study area is located in an urban area.*

AUTHORS: The existing PWS-network is tested on the demands specified in the literature for urban hydrological applications. The required spatial and temporal resolutions of measurements are considered. Based on the number of stations and frequency of data upload, the resolution approached the required values stipulated in literature. However, there were errors in the datasets originating from data transfer as well as other sources. These errors can be compensated by averaging rainfall measurements in space and time. This decreases the effective resolutions of the network.

It shall be noted more clearly that this study focused on whether this rainfall data source meets the requirements of urban rainfall monitoring as specified by previous research (see Introduction) rather than of reexamining these requirements.

References

Ciach, G. J. and Krajewski, W. F.: Analysis and modeling of spatial correlation structure in small-scale rainfall in Central Oklahoma, Advances in Water Resources, 29, 1450–1463, 2006.

Habib, E., Krajewski, W. F., and Ciach, G. J.: Estimation of rainfall interstation correlation, Journal of Hydrometeorology, 2, 621–629, 2001.

Krajewski, W. F., Ciach, G. J., and Habib, E.: An analysis of small-scale rainfall variability in different climatic regimes, Hydrological Sciences Journal, 48, 151–162, 2003.

Overeem, A., Buishand, T. A., and Holleman, I.: Extreme rainfall analysis and estimation of depth-duration-frequency curves using weather radar, Water Resources Research, 45, 2009a.

Overeem, A., Holleman, I., and Buishand, A.: Derivation of a 10-year radar-based climatology of rainfall, Journal of Applied Meteorology and Climatology, 48, 1448–1463, 2009b.

Overeem, A., Leijnse, H., and Uijlenhoet, R.: Measuring urban rainfall using microwave links from commercial cellular communication networks, Water Resources Research, 47, 2011.

Peleg, N., Ben-Asher, M., and Morin, E.: Radar subpixel-scale rainfall variability and uncertainty: lessons learned from observations of a dense rain-gauge network, Hydrology and Earth System Sciences, 17, 2195–2208, 2013.

Tokay, A. and Öztürk, K.: An experimental study of the small-scale variability of rainfall, Journal of Hydrometeorology, 13, 351–365, 2012.

Villarini, G., Mandapaka, P. V., Krajewski,W. F., and Moore, R. J.: Rainfall and sampling uncertainties: A rain gauge perspective, Journal of Geophysical Research: Atmospheres, 113, 2008.

---

## Author Response (AR1)

**Author's response**

December 18, 2016

5   The discussion paper has been revised based on the comments from the reviewers and the editor. This document lists those comments, each followed by the response of the authors. This is followed by a marked-up manuscript, indicating the changes between the previous and last version of the paper. Revisions are indicated with color, where deleted and added text is indicated with respectively red and blue.

Page and line numbers in the author's response refer to those in the marked-up version of the manuscript in this document.

10  Page and line numbers from the original comments referring to the previous manuscript have not been altered.

*REVIEWER 1: The paper discusses the advantages of crowdsourced weather station data (rainfall measurement) to obtain rainfall information suitable for hydrology studies in urban areas, i.e., rainfall measurements that the need to have high temporal and spatial resolutions. The paper is, to the best of my knowledge, the first attempt to quantify the errors of rainfall data made available from local, distributed and crowdsourced weather stations, which makes it an interesting study. In the paper the crowdsourced rainfall data are compared with dedicated rain gauges and rainfall radar data as these are the common rainfall data sources used in urban hydrology.*

AUTHORS: We thank the reviewer for the valuable review of this paper. We appreciate the constructive comments, and address each of them below.

*REVIEWER 1: Here some suggestions:*

*(1) Some sentences are too vague and need to be rephrased to convey a clear message: e.g., what do authors mean by "... return time of less than a few years"?*

AUTHORS: The section discussed here was intended to highlight the limitations of the filter that was used on the Amsterdam dataset. "a return time of less than a few years" refers to the return times of Dutch rainfall amounts found by Buishand and Velds (1980), Overeem et al. (2008) and Buishand and Wijngaard (2007), as documented by the latter.

The filter that is discussed in this section contains a dynamic maximum threshold, which filters out intervals with values higher than 50 mm h$^{-1}$ above the median rainfall intensity of the surrounding stations. For example, in case of 10 min time series the filter works as follows; If for a certain 10 min interval the median rain intensity of surrounding stations is 4 mm h$^{-1}$, the upper threshold for rainfall intensity becomes 54 mm h$^{-1}$ for this particular interval. According to the values found in the research, this corresponds with a return time between 1 and 2 years (Buishand and Velds, 1980; Buishand and Wijngaard, 2007).

As these return time values are different for other interval lengths and median values of surrounding stations, the statement was phrased in a general way.

In conclusion, realistic rainfall measurements could be excluded by the filter. We would like to stress that visual comparison with gauge adjusted radar data indicates that this was not the case for the dataset used is this paper. However, for operational application of a filter with an upper threshold, this should be taken into consideration.

Other sentences throughout the manuscript have been adjusted to make their message more clear.

*(2) The structure of the manuscript deserves to be revised. See for example: (a) the order figure numbers appear in the manuscript is cumbersome and makes the manuscript confusing (why Fig. 5 appears earlier than Fig. 3?)*

AUTHORS: In Sect. 2.1.3 the experimental set-up is described, and the reader is referred to the inset in Fig. 5 where the set-up is visualized. This reference precedes references to Fig. 3 and Fig. 4, as the results that are also visualized in Fig. 5 are discussed in Sect. 3.1. The authors acknowledge this may be cumbersome to the reader, though prefer to not split up Fig. 5 in two figures.

*(3) Lines 18-24 in Page 12 are not conclusions. The authors may want to move these sentences to another section of the manuscript. Also, lines 29-33 page 12 are not conclusions.*

AUTHORS: The sections that did not belong in the Conclusions section have been removed.

*(4) Figure 5 does not show "a dedicated experimental set-up" (page 6, lines 2-3), i.e. the text does not match what is seen in the Figure. The authors may want to adjust the text of the Figure.*

AUTHORS: With this sentence the authors refer to the photo in Fig. 5 that shows the experimental set-up of the rain gauges. In page 6, line 22, the reference has been changed to "Fig. 5, photo inset".

*(5) Figure 1. The "black dot" KNMI radar product is not visible in the plot (only in the legend). Authors may want to adjust the plot /or legend).*

AUTHORS: The symbols in Fig. 1 have been adjusted.

*(6) There is room for improving the English language; incomplete sentences (e.g., Page 5, line 6) and minor typos (e.g., "criterium" should read "criterion" in Page 11, line 19) can be found in the manuscript; "CV" is only defined in Fig. 10 legend.*

AUTHORS: The overall text has been reevaluated for English language.

*REVIEWER 2: Summary*

*This study discusses a new technique for measuring rainfall. This consequently would satisfy the need for hydrological analyses like urban hydrology, where high temporal and spatial resolution rainfall data are required. Although several studies have addressed alternative ways for measuring rainfall such as using microwave links, this study specifically investigates rainfall information from automatic personal weather stations (PWSs) for point-measurement purposes. I found the topic quite interesting as such approaches may indeed provide valuable information.*

AUTHORS: We want to thank the reviewer for the review of this paper. We appreciate all the suggestions to improve this manuscript, and address each of them below.

*REVIEWER 2: General comments:*

*I found the manuscript a bit hard to follow. The authors may consider the following points for improvements. The structure of the manuscript may need some modifications. For example, you may introduce the "Methodology" the "Data and study area" in separated sections, and not combined. Providing some important details would make getting the message of the manuscript easier. More information is required explaining the functionality of the devices used in this study. Although some devices do not have the sought information, providing available information, e.g. for Netatmo, would help to better follow the text. You may even add some figures for that reason. The message of some sentences is not clear. Furthermore, some sentences have minor/major language problems and/or they are too long. You may consider English proofreading. Another point is the objective of this study. The last paragraph in introduction addresses the objective of this study. You may clearly add the fact that using these measurements is economically reasonable comparing to conventional techniques. At the end I would like to add*

*that there are some facts given without proper references.*

AUTHORS: We thank the reviewer for the suggestions on how to improve the manuscript. We havel reevaluated the structure of the sections in order to improve readability. It was preferred to not split up the Methodology section, and keep the division between data collection and data analysis. The text has been scanned for sentences that are unclear and/or without proper reference. The argument of the reviewer regarding the objective of this study has been added to the last paragraph of the Introduction (page 4, lines 21 -23).

Section 2.1.1. contains product specifications (as provided by the manufacturer) on the Netatmo rain gauge devices. More information has been provided on the data transfer between devices in a station; the rain module, i.e. the tipping gauge, communicates in a wireless manner to the indoor module over distances up to 100 m. Every 5 minutes (time step may vary) the number of tips in that interval is communicated from the indoor module to the online dashboard via a WiFi-connection. (page 5, lines 8 - 12)

*REVIEWER 2: Scientific comments*:

*You addressed a study where they used "average value of 12 pixels", and furthermore, you addressed the difficulties radar data have, especially when comparing to "groundbased measurements". You used in this study 1 radar pixel as the reference at the end. You may elucidate the reason for that.*

AUTHORS: Though radar provides rainfall estimates with a large coverage, they are indirect measurements of rainfall averages over a spatial area (pixels 0.92 km$^2$ for our C-band Doppler radar product) representative for a significant altitude above the ground (~1.5 km). Because radar measures in a volume aloft, there will always be differences with point measurements at the ground. This difference will not decrease by averaging rainfall in multiple radar pixels. Although averaging over radar pixels may smooth such an error, it also implies that rainfall from a much larger area is compared to a point measurement, which will introduce an additional difference. Figure 5 shows the scatter plots of Netatmo measurements with radar data as well as with the pit gauge reference. This figure is meant to highlight that even though the Netatmo rain gauges show good correspondence with the pit gauge reference (both are point measurements), a larger spread is found when comparing with gauge-adjusted radar rainfall. This is helpful in the interpretation of the analyses of PWS rainfall data in Amsterdam. For these time-series no ground-based point reference is available, which is why we compare with the gauge-adjusted radar data.

*REVIEWER 2: Also, you may explain the method (you) used for correcting (adjusting) radar data.*

AUTHORS: Section 2.1.2. describes the radar dataset used in this research. Data from two C-band Doppler weather radars in De Bilt and Den Helder have been adjusted with ground-based rainfall measurements. This adjustment was already done and has not been performed as part of this study. The dataset has been made available by KNMI. For information on the methods used we refer more clearly to previous research (Overeem et al., 2009a, b, 2011) in Sec. 2.1.2 (page 6, line 15).

*REVIEWER 2: The same is valid for the experimental set-up. You may add how exactly the reference device "KNMI pit gauge" measures rainfall.*

AUTHORS: The KNMI pit gauge has been introduced in Sect. 2.1.3., which also includes information on the pit gauge configuration. It is an electronic rain gauge, measuring cumulative rainfall every 12 s. A potentiometer connected to a floating unit measures the amount of liquid water. A heating element makes it possible for the device to accurately measure solid precipitation.

*REVIEWER 2: I could not really understand the reason why you used Pearson correlation coefficient for your analyses. You may add some other criteria like bias, root mean square error, etc.*

AUTHORS: The Pearson correlation coefficient is a widely used statistic to describe correspondence between rain gauge measurements, in order to characterize the complex spatial structure of rainfall patterns. The dependence of the Pearson correlation coefficient with distance between gauges has been evaluated Habib et al. (2001), Krajewski et al. (2003), Ciach and Krajewski (2006), Villarini et al. (2008), Tokay and Öztürk (2012) and Peleg et al. (2013). In order to compare with results in these papers, a similar method was chosen.

Figure 6 describes the accumulation interval dependence of the correlation between gauges and the pit gauge reference. The same figure has been made for the coefficient of variation ($CV$) of the residuals. The $CV$ values decrease with increasing accumulation interval where they approach zero (indicating a perfect match) at 15 minute intervals. The values of the Wunderground timeseries approach zero at a slower pace than the Netatmo original data. As this $CV$-analysis did not provide any additional information beyond that communicated by Fig. 6, it has not been included in the manuscript.

The Pearson correlation coefficient and the $CV$ give information on the random error. Figure 5 and Fig. 7 give information on the bias.

*REVIEWER 2: Regarding the "inter-station correlation", you may add the way you estimated the parameters.*

AUTHORS: The parameters in Fig. 12 have been determined by fitting Eq. (2), where the correlations between stations and the inter-gauge distances were used as input. Fitting was done by determining the nonlinear (weighted) least-squares estimates of the parameters. This will be included in Sect. 3.4 (page 11, lines 14-15).

*REVIEWER 2: Some of the specific comments for the abstract and introduction section is provided in the following:*

*P.1 - L.7: are you referring by "63 stations" to the Netatmo network in Amsterdam?*

AUTHORS: The 63 stations do not exclusively refer to Netatmo stations. Figure 2 shows which of these stations are of type Netatmo (square symbols) and which are of other types (triangle symbols). The time-series are obtained from the Wundermap website, as described in detail in Sect 2.1.1.

*REVIEWER 2: P.1 – L.10: "the method of data transfer to the online platform causes considerable errors in the datasets obtained." This phrase may need some modifications.*

AUTHORS: The sentence will be rephrased in "The sensor performance in the experimental set-up and the density of the PWS-network are promising. However, features in the online platforms cause changes in the time-series, resulting in considerable errors in the datasets obtained." (page 1, lines 9-11)

*REVIEWER 2: P.1 – L.16: Does "data conversion" refer to "data transfer" you provided some lines earlier? The word "conversion" sounds a bit strange in this contest.*

AUTHORS: This refers to adjustments made to the measurement time-series by features of the online platforms. Data conversion occurs within the data transfer. This has been rephrased in order to avoid confusion.

*REVIEWER 2: P.2 – L.3: Why "Rainfall" starts with capital letter? And "from the radar"; you may change to "from the radar station".*

AUTHORS: This should indeed not be a capital. "from the radar" was changed into "from the radar station" as suggested (page 2, line 7).

*REVIEWER 2: P.2 – L.4: "and which may.." I suppose one may not put "and" and "which" together.*

AUTHORS: This has been corrected.

*REVIEWER 2: P.2 – L.7: Does "spatial resolution" mean "network density"?*

AUTHORS: Spatial resolution is directly related to network density, only if all stations in the network are measuring constantly. In most PWS-networks, spatial resolution is lower than the actual network density because of station outages and periods of fewer measurements in time.

*REVIEWER 2: P.2 – L.10: You may add a comma after "Hydrological models" and a comma before "have minimized".*

AUTHORS: This has been corrected.

*REVIEWER 2: P.2 – L.15: What is the spatial resolution of 3 km? Is it a 3 km x 3 km?*

AUTHORS: A spatial resolution of 3 km generally corresponds with a 3 km x 3 km square. Here it means that measurements are taken on average 3 km apart. For crowdsourced measurements, the inter-gauge distance is likely to vary slightly, as the placement of stations is irregular and not in a regular grid.

*REVIEWER 2: P.2 – L.21: Bruni et al. (2015) "addressed".*

AUTHORS: All the tenses in the document regarding such references to literature have been changed.

*REVIEWER 2: P.2 – L.24: When explaining "Smearing"; "the ratio of the resolution": are you referring to temporal, spatial or tempo-spatial resolution?*

AUTHORS: Here we refer to spatial resolution ratios. This has been clarified in the text, as well as the highest resolution of the radar data (page 2, lines 25-31).

*REVIEWER 2: P.2 – L.30: Ochoa-Rodriguez et al. (2015) "evaluated the"*

AUTHORS: This has been corrected.

*REVIEWER 2: P.2 – L.31: You may add a comma after "1 km" and a comma before "was found".*

AUTHORS: This has been corrected.

*REVIEWER 2: P.2 – L.34: Not all radar products are in 5 min temporal resolutions. Some X-Band radars work in 30-sec temporal resolution.*

AUTHORS: The phrase will be changed to "Temporal resolutions should ideally be below the 5 min intervals currently available in most operational weather radar-products," (page 3, line 5).

*REVIEWER 2: P.2 – L.35: "impact on the accuracy than coarsening spatial scales does." I would change the word scale to "resolution". Furthermore, you may omit "does" at the end of the sentence.*

AUTHORS: This has been corrected (page 3, lines 7-8).

*REVIEWER 2: P.3 – L.2: "evaluate" to "evaluated". You may rephrase the entire sentence.*

AUTHORS: This has been corrected.

*REVIEWER 2: P.3 – L.9: What type of sensors are you referring to? Rain gauges?*

AUTHORS: The sensors we refer to are any type of sensors that measure rainfall. We will clarify this in our revised manuscript. This could be rain gauges, though it could also refer to disdrometers (page 3, line 17).

*REVIEWER 2: P.4 – L.10: But you did not investigate for any urban applications. Your study area is located in an urban area.*

AUTHORS: The existing PWS-network is tested on the demands specified in the literature for urban hydrological applications. The required spatial and temporal resolutions of measurements are considered. Based on the number of stations and frequency of data upload, the resolution approached the required values stipulated in literature. However, there were errors in the datasets originating from data transfer as well as other sources. These errors can be compensated by averaging rainfall measurements in space and time. This decreases the effective resolutions of the network.

*ADDITIONAL AUTHOR'S COMMENTS:*

In the previous version of the manuscript some conclusions were made regarding processing errors in obtaining data via the Netatmo Weathermap platform, that later proved to be due to a faulty API. Also, it was found that the original Netatmo time-series were not only available in real-time via this platform, but also as historic data. The conclusions in the manuscript have been adapted according to this new information, in the last alinea of section 3.1 (page 9, lines 13-34), and the Conclusions section (page 15, lines 1-5).

Figure 4 was made with an analysis that included some faulty assumptions. The figure has been corrected, this time with the original unfiltered data. In order to avoid confusion the points that indicate on which dt and dx combinations the fraction calculation has been performed were excluded. The overall conclusions from the figure have not been changed dramatically, though now they make more sense as they include some temporal dependence at short time steps.

Figure 12 has been made, with thicker dotted lines to improve readability.

**References**

Buishand, T. A. and Velds, C. A.: Neerslag en verdamping [Precipitation and evaporation], Royal Netherlands Meteorologic Institute, 1980.

[revised manuscript text omitted]